# Heart Rate Variability and Long Chain *n*-3 Polyunsaturated Fatty Acids in Chronic Kidney Disease Patients on Haemodialysis: A Cross-Sectional Pilot Study

**DOI:** 10.3390/nu13072453

**Published:** 2021-07-18

**Authors:** Ana M Pinto, Helen L MacLaughlin, Wendy L Hall

**Affiliations:** 1Department of Nutritional Sciences, School of Life Course Sciences, King’s College London, 150 Stamford Street, London SE1 9NH, UK; ana_margarida.pinto@kcl.ac.uk; 2School of Exercise & Nutrition Sciences, Queensland University of Technology, Brisbane, QLD 4059, Australia; h.maclaughlin@qut.edu.au

**Keywords:** haemodialysis, chronic kidney disease, heart rate variability, autonomic dysfunction, omega-3 PUFA

## Abstract

Low heart rate variability (HRV) is independently associated with increased risk of sudden cardiac death (SCD) and all cardiac death in haemodialysis patients. Long chain *n*-3 polyunsaturated fatty acids (LC *n*-3 PUFA) may exert anti-arrhythmic effects. This study aimed to investigate relationships between dialysis, sleep and 24 h HRV and LC *n*-3 PUFA status in patients who have recently commenced haemodialysis. A cross-sectional study was conducted in adults aged 40–80 with chronic kidney disease (CKD) stage 5 (n = 45, mean age 58, SD 9, 20 females and 25 males, 39% with type 2 diabetes). Pre-dialysis blood samples were taken to measure erythrocyte and plasma fatty acid composition (wt % fatty acids). Mean erythrocyte omega-3 index was not associated with HRV following adjustment for age, BMI and use of β-blocker medication. Higher ratios of erythrocyte eicosapentaenoic acid (EPA) to docosahexaenoic acid (DHA) were associated with lower 24 h vagally-mediated beat-to-beat HRV parameters. Higher plasma EPA and docosapentaenoic acid (DPA*n*-3) were also associated with lower sleep-time and 24 h beat-to-beat variability. In contrast, higher plasma EPA was significantly related to higher overall and longer phase components of 24 h HRV. Further investigation is required to investigate whether patients commencing haemodialysis may have compromised conversion of EPA to DHA, which may impair vagally-mediated regulation of cardiac autonomic function, increasing risk of SCD.

## 1. Introduction

Cardiovascular disease is particularly prevalent in chronic kidney disease (CKD) [1], and accounts for 43% of all-cause mortality among dialysis patients [2], compared to an estimated 11% of deaths in the general population [3]. Risk of sudden cardiac death (SCD) is doubled when a patient with CKD stage 5 (kidney failure) starts dialysis [4] and in patients on haemodialysis, SCD accounts for two thirds of all cardiac deaths and one fourth of all-cause mortality [2]. Patients with kidney failure are a population with low heart rate variability (HRV) indicating autonomic dysfunction [5]. Low HRV has been independently associated with increased risk of all-cause mortality, SCD and all cardiac death in haemodialysis patients [6,7,8]. Patients on haemodialysis have lower serum long chain *n*-3 polyunsaturated fatty acids (LC *n*-3 PUFA) levels compared to control populations without CKD [9,10], and lower serum LC *n*-3 PUFA: arachidonic acid (AA) ratios were predictive of CVD events [10]. Fish consumption has been associated with reduced mortality in an observational study with incident dialysis patients [11], and Friedman et al. reported an inverse association between erythrocyte LC *n*-3 PUFA levels and mortality in haemodialysis patients [12]. Supplementation with LC *n*-3 PUFA is a promising candidate for SCD prevention in this population due to their potential anti-inflammatory, anti-oxidative and anti-arrhythmic effects on the cardiac myocytes [13,14,15]. Mixed results have been obtained in previous investigations into the effect of LC *n*-3 PUFA on HRV in healthy subjects but there is stronger evidence in myocardial infarction survivors of benefits from fish oil supplementation [16]. Few studies have investigated the relationship between LC *n*-3 PUFA and HRV in CKD patients [17,18]. Identifying positive associations between tissue LC *n*-3 PUFA status and HRV in patients with kidney failure would provide important evidence to support the potential application of LC *n*-3 PUFA supplementation to improve cardiac autonomic function and help prevent SCD.

It is hypothesised that the proportion of eicosapaentanoic acid + docosahexaenoic acid (EPA + DHA) in the erythrocyte will be independently positively associated with 24 h HRV in patients with CKD stage 5 who have recently commenced haemodialysis. This study aimed to describe LC *n*-3 PUFA status in patients who recently commenced haemodialysis, and to investigate relationships between erythrocyte membrane LC *n*-3 PUFA composition and HRV in these patients, including comparison to a reference healthy population. Associations between plasma LC *n*-3 PUFA composition and HRV will also be explored.

## 2. Materials and Methods

This was a cross-sectional pilot study with ethical approval obtained from NRES Committee London–Camberwell St Giles (REC ref: 14/LO/0186). The study was registered on clinicaltrials.gov (NCT02014792) and conducted at King’s College Hospital (KCH) and Guy’s and St Thomas’ Hospital (GSTT) and their 10 respective satellite dialysis units in the London area. All subjects gave their informed consent for inclusion before they participated in the study. The study was conducted between July 2014 and January 2016, in accordance with the rules of the Declaration of Helsinki of 1975, revised in 2013.

The primary dependent variables were measures of 24 h overall HRV during haemodialysis: SDNN and triangular index (defined below). The primary independent variable was the proportion of EPA + DHA (Omega-3 Index, O3I; %) in the erythrocyte membrane. Secondary dependent outcome variables included other time- and frequency-domain, and non-linear HRV parameters, during 2 h dialysis, 24 h and sleep-time. Secondary independent outcome variables included erythrocyte and plasma proportions (%) of EPA, docosapentaenoic acid (DPA*n*-3) and DHA. Other descriptive variables included background dietary intakes (using the EPIC-Norfolk food frequency questionnaire; data was analysed using an in-house customized Microsoft Access database and exported to Microsoft Excel for further processing. The individual LC *n*-3 PUFA intake calculations were determined using the nutrient composition values from the McCance and Widdowson’s ‘The Composition of Foods Integrated Dataset’ with additional data obtained from manufacturers and USDA values when required.) and clinical risk factors, including history of sleep apnoea or suspected sleep apnoea (using Epworth Sleepiness Scale and Berlin Questionnaire), waist circumference, 12 month average SBP and DBP and BP variability prior to starting haemodialysis, hs-C reactive protein (CRP) at start of dialysis session, serum creatinine, albumin, sodium, potassium, phosphate and calcium before and after the dialysis session.

The inclusion criteria were: women and men with CKD stage 5 commencing haemodialysis (6 to 10 weeks after the first session), aged 40–80 years and able to provide written informed consent. The exclusion criteria included: unwilling to participate, history of chronic liver disease or neuropathy, infection or antibiotics within the last month and history of prior renal replacement therapy including transplantation.

Participants were patients under the care of a nephrologist. They were identified through the course of usual clinical care, or via electronic database search conducted by the project dietitian and nephrologist at King’s College Hospital NHS Foundation Trust and Guy’s and St Thomas’ NHS Foundation Trust, respectively. Patients had haemodialysis treatment on three non-consecutive days a week. Eligible patients enrolled in the study 6 to 10 weeks after commencing haemodialysis treatment and two study days, at least 1 week apart, were scheduled to yield duplicate data. Study days were not scheduled right after the two consecutive days of intradialytic period, also known as the “long gap”, to avoid even greater elevations in uraemia and to maintain consistency across duplicate study measurements.

In both study visits, pre- and post-dialysis blood sample were taken by a nurse (Appendix A). Waist circumference was taken on patient’s arrival on the first study visit before going on the dialysis machine. Immediately after the nurse started the patient on haemodialysis treatment, the researcher fitted the monitor to record HR and HRV for a 24 h period. Additional information was obtained from the medical records. All patients continued with the best medical care, as appropriate, at the nephrologist’s discretion.

Heart rate variability was measured using the Actiheart monitoring equipment (CamNtech Ltd., Cambridge, UK). Data processing of the HRV recordings was carried out using the Actiheart software (version 4.0.91, CamNtech Ltd., Cambridge, UK) and Kubios HRV analysis software (Biosignal Analysis and Medical Imaging Group, Department of Physics, University of Kuopio, Finland) where all artefacts were excluded from the analysis. Patients were given an activity and event diary to complete with information about the duration of dialysis, type of activities done, and nap and sleep times at night. HRV and HR/interbeat interval (IBI) data were analysed for a standardised 2 h of haemodialysis (from the moment the monitors were fitted), for 24 h (in order to be considered valid and used for analysis a minimum of 18 h of the recording had to be analysable) and sleep time. HRV outcomes included time and frequency-domain parameters; time-domain parameters are based on the time intervals between adjacent QRS (Q, R and S being points on the R wave seen on an ECG during ventricular depolarisation, and R being the peak upward deflection) complexes (normal-to-normal (NN) intervals) whereas frequency-domain parameters employ power spectral analysis of NN intervals to determine the power (variance) within frequency bands [19]. Time-domain parameters included SDNN, standard deviation of the average normal-to-normal intervals in 5 min segments of the whole recording (SDANN), square root of the mean of the sum of the squares of differences between adjacent NN intervals (RMSSD), the percentage of adjacent normal-to-normal intervals that differed by >50 ms (pNN50) and triangular index (TI), the integral of the density distribution (the number of all NN intervals) divided by the maximum of the density distribution. Frequency-domain parameters included high-frequency (HF), low-frequency (LF) and very-low-frequency (VLF) power, and the ratio of the LF and HF band powers (LF:HF). A non-linear parameter using Poincaré plots of short-term variability (SD1) against long-term variability (SD2) was also calculated as a measure of complexity of HRV distribution over the duration of the recording. SDNN, LF and TI represent overall variability. Short-term (beat-to-beat) components of HRV include RMSSD, pNN50 and HF. SDANN and VLF reflect longer-phase components of variability.

Pre- and post-dialysis blood samples collected in SST^TM^ tubes (BD Vacutainer^TM^, Franklin Lakes, NJ, USA) were used to determine serum sodium, potassium, phosphate and calcium as well as albumin and creatinine before and after dialysis and hs-CRP before dialysis only. These analyses were performed by a clinical pathology accredited biochemistry laboratory (ViaPath, Kings College Hospital). Pre-dialysis blood samples collected in EDTA tubes were used to determine erythrocyte membrane fatty acid composition (% weight) and plasma fatty acid composition (% weight) at King’s College London. Proportions of fatty acids in whole plasma and erythrocyte membranes were analysed by GC (Agilent 7890A GC; Agilent Technologies, Santa Clara, CA, USA) with a BPX70 GC column (length 25 m, internal diameter 0·32 mm, film thickness 0·25 μm) custom designed for separation of fatty acid methyl esters (SGE Analytic Science, Ringwood Victoria, Australia) following transesterification, as previously described [20], but substituting toluene for benzene and using pentadecanoic acid as an internal standard. The O3I was defined as the sum of % weight EPA + DHA in erythrocytes.

Statistical analyses were carried out using IBM SPSS Statistics 21.0 (Statistical Product and Service Solutions; IBM Corp., Armonk, NY, USA) and statistically significance was considered at *p*-value < 0.05. Values from first and second visit were averaged in all subjects. Simple linear regression was used to analyse relationships between LC *n*-3 PUFA status (O3I, erythrocyte and plasma EPA, DHA and DPA, and erythrocyte EPA:DHA ratio) and HRV variables. Multiple linear regression was used to control for confounding factors, age, BMI and use of β-blockers, which are all related to HRV. Non-normally distributed data were normalised by natural logarithm (LN) before performing the analysis. Standardised residuals were checked for normality. Chi-square (χ 2) test and independent *t*-tests were used to compare categorical and continuous variables, respectively, between haemodialysis patients and a healthy cohort. Non-normally distributed data were normalised by LN transformation (results shown as geometric means and 95% CI) before analysis by independent *t*-test. If LN transformation failed to yield a normal distribution, a Mann–Whitney U test was applied to compare groups (results shown as medians with lower and upper quartiles).

## 3. Results

### 3.1. Patient Characteristics

A total of 49 patients were enrolled, 45 patients completed the first study visit and 31 completed both study visits (Figure 1). Of the 45 patients, 39 patients provided HRV recordings of sufficient quality to be included in the 2 h dialysis period, 40 provided sleep-time HRV recordings of sufficient quality (minimum 3 h period), and 35 provided 24 h recordings of sufficient quality for analysis (minimum 18 h period). There was only one patient where it was not possible to obtain erythrocyte membrane fatty acid composition due to technical issues.

Demographic and clinical characteristics for the total population enrolled in the study is presented in Table 1, including 2 h HRV parameters during dialysis. Most patients presented hypertension (90.8%) and the second most common co-morbidity was type 2 diabetes (38.6%). Nearly half the patients were taking statins, β-blockers, erythropoietin therapy and iron injections, and more than half were taking vitamin D supplements. Protein, fat and carbohydrate made up 18 (5), 36 (8) and 45 (7)% (medians with IQR) of energy intakes respectively. Median total sugar intake was 75 g/d (18% (7) energy intake) and mean saturated fatty acids (SFA) intakes were 13% (SD 3).

### 3.2. LC n-3 PUFA and 2 h Dialysis, Sleep-Time and 24 h HRV

For participants with complete duplicate visits there were significant moderate to strong correlations between visits 1 and 2 for all HRV parameters: dialysis, n = 23, 0.578 < r < 0.880, *p* < 0.01; sleep time, n = 24, 0.468 < r < 0.913, *p* < 0.05; and 24h, n = 19, 0.630 < r < 0.812, *p* < 0.01. Correlations between study visits 1 and 2 for LC *n*-3 PUFA were also moderate to strong: erythrocyte EPA, r = 0.687, *p* < 0.001; erythrocyte DHA, r = 0.497, *p* = 0.010; and erythrocyte O3I, r = 0.538, *p* = 0.005 (all n = 26); plasma EPA, r = 0.729, *p* < 0.001; plasma DPA(*n*-3), r = 0.907, *p* < 0.001; and DHA, r = 0.885, *p* < 0.001 (all n = 31).

The median [lower and upper quartile] erythrocyte O3I of this population was very low (3.3%; 2.5, 3.8); only 2 individuals had O3I > 5%. Median erythrocyte EPA [lower and upper quartile] was 0.7% [0.5, 0.9] and DHA was 2.5% [2.0, 3.2]. Median plasma EPA was 0.8% [0.5, 1.2], DPA(*n*-3) was 0.4% [0.4, 0.5], and DHA was 1.7% [1.3, 2.2]. There was a strong significant correlation between plasma and erythrocyte EPA (r = 0.704; *p* < 0.001), and between plasma and erythrocyte DHA (r = 0.608; *p* < 0.001). No correlations were found between plasma or erythrocyte LC *n*-3 PUFA proportions and estimated PUFA, EPA, DHA or oily fish intake assessed by FFQ.

Mean 24 h HR was 83 bpm (SD 10.9) and IBI was 747 ms (SD 131). Twenty-four hour overall HRV was low: 84 ms (SD 26) for SDNN and median Ti was 17 [lower and upper quartiles 15 and 27]. Mean and median parameters of long-phase and short-phase variability are presented in Appendix B.

### 3.3. LC n-3 PUFA and HRV in Haemodialysis vs. Reference Healthy Population

A paired sub-group of a healthy cohort from a previously reported randomized controlled trial, the MARINA study [21], were used as a reference population. Individuals from the current haemodialysis cohort were pair-matched by age, sex and BMI with individuals from the healthy reference population. Table 2 displays the results comparing LC *n*-3 PUFA intake and blood composition, hs-CRP, and HRV parameters during sleep time. Compared to the healthy cohort, patients in the haemodialysis cohort had significantly higher sleep-time HR and lower HRV, higher hsCRP, as well as significantly lower proportions of erythrocyte membrane and plasma LC *n*-3 PUFA, and higher erythrocyte EPA:DHA ratio, whilst EPA and DHA dietary intakes were not significantly different. There were other notable differences in erythrocyte fatty acid composition, with significantly higher proportions of MUFAs, *cis*-vaccenic acid (18:1*n*-7), oleic acid (18:1*n*-9), and palmitoleic acid (16:1*n*-7); SFAs, stearic acid (18:0) and palmitic acid (16:0), and the erythrocyte palmitic acid (PA):linoleic acid (LA, 18:2*n*-6) ratio in patients in the haemodialysis cohort compared to the healthy reference group. Furthermore, there were significantly lower proportions of erythrocyte n-6 PUFAs, docosapentaenoic acid (DPA; 22:5*n*-6), arachidonic acid (20:4*n*-6), dihomo-γ-linolenic acid (DGLA; 20:3n-6), and linoleic acid (18:2*n*-6) in patients in the haemodialysis cohort compared to the healthy reference group.

### 3.4. Associations between LC n-3 PUFA and HRV in Haemodialysis Patients

#### 3.4.1. Erythrocyte Membrane LC *n*-3 PUFA

No significant associations were observed between erythrocyte O3I, EPA or DHA (%) and HRV parameters during dialysis, sleep nor 24 h following adjustment (Table 3). EPA:DHA ratios were inversely associated with RMSSD (adjusted standardised β = −0.377, *p* = 0.044), LF power (adjusted standardised β = −0.289, *p* = 0.043), HF power (adjusted standardised β = −0.436, *p* = 0.008), and SD1:SD2 (adjusted standardised β = −0.364, *p* = 0.047).

#### 3.4.2. Plasma LC *n*-3 PUFA

Plasma DHA (%) was positively associated with Ti during dialysis (adjusted standardised β = 0.331, *p* = 0.022) (Table 4) but no significant associations were observed between plasma EPA or DPA and any of the HRV parameters during dialysis.

During sleep, DPA(*n*-3) was negatively associated with pNN50 (adjusted standardised β = −0.321, *p* = 0.041) and SD1:SD2 (adjusted standardised β = −0.330, *p* = 0.045), both indicators of short-term variability, and EPA was also negatively associated with SD1:SD2 (adjusted standardised β = −0.354, *p* = 0.036). DHA was not related to sleep-time HRV, nor 24 h HRV.

Significant moderate positive associations were observed between plasma EPA and longer phase components of 24 h HRV (SDANN adjusted standardised β = 0.405, *p* = 0.016; VLF power adjusted standardised β = 0.481, *p* = 0.004) as well as overall 24 h HRV (SDNN adjusted standardised β = 0.441, *p* = 0.009). A moderate significant negative association between EPA and 24 h SD1:SD2 was observed (adjusted standardised β = −0.482, *p* = 0.004), and between plasma DPA(*n*-3) and 24 h short-term variability indicators RMSSD (adjusted standardised β = −0.387, *p* = 0.024), and pNN50 (adjusted standardised β = −0.394, *p* = 0.021).

## 4. Discussion

Arrhythmias and cardiac arrests account for over a quarter of deaths among haemodialysis patients [22] and the first three months following commencement of haemodialysis treatment are particularly high-risk for SCD due to factors relating to the parameters of the dialysis itself that may trigger arrhythmia, but also the high prevalence of pre-existing cardiovascular disease and diabetes [23]. This study set out to characterise erythrocyte membrane and plasma LC *n*-3 PUFA in a UK patient population who have recently commenced haemodialysis, as well as exploring relationships with HRV parameters during this critical period in their treatment. Previous reports on haemodialysis patients in other countries (Italy, Denmark, USA, Germany and Serbia) have shown lower EPA, DHA, or both EPA + DHA, in erythrocyte membranes and/or plasma compared to healthy controls or healthy populations [9,24,25,26,27,28]. Shoji et al. showed greater levels of EPA and DHA in plasma in haemodialysis patients compared to healthy controls in Japan [10] and Friedman et al. reported a greater erythrocyte DHA and O3I, but lower plasma DHA, in haemodialysis patients (n = 75) compared to a healthy control group (n = 25) in the USA [29]. Compared to other haemodialysis cohorts, the mean levels of DHA and O3I in the present study population were amongst the lowest, being 61% and 57% lower, respectively, than a matched healthy cohort.

Erythrocyte membrane fatty acid composition is considered to be a better marker of cardiovascular disease risk than plasma, as erythrocyte membrane phospholipids are more reflective of the cardiomyocytes fatty acid profiles [30]. None of the patients in this study presented an O3I above 8% which has been associated with the greatest cardio-protection; mean O3I was below 4%, which has been associated with the least cardio-protection [31]. Background diet may play a role, but this was not supported by food frequency questionnaire data on self-reported fish intake. It is most likely that the major contributing factor to low erythrocyte LC *n*-3 PUFA content was related to the pathology of kidney disease. Anaemia is prevalent in stage 5 CKD due to reduced production of erythropoietin, a hormone produced by the kidney that regulates erythrocyte production from bone marrow (necessitating erythropoietin therapy in most haemodialysis patients), and also as a result of uraemic inhibitors of erythrocyte production [32]. Other contributors to anaemia include reduced lifespan of erythrocytes due to oxidative stress, inflammation and haemolysis, which may be a limitation in using erythrocyte membrane fatty acid composition as a proxy marker of tissues of interest such as cardiac myocytes or neurons. Thus, although the utility of erythrocyte fatty acid composition as a biomarker of moderate-long term dietary intake has been established in other populations [33], the same relationships with dietary intake and tissue status cannot be assumed in populations in advanced stages of CKD. These relationships are likely to be further disrupted in patients with kidney failure due to haemodialysis-related factors such as chronic inflammation, oxidative stress and altered lipid metabolism/uptake into membranes. Our data show that variability in erythrocyte LC *n*-3 PUFA contents is indeed limited to a small range, and it is yet to be established whether this is an accurate representation of general tissue status in haemodialysis patients compared to other sample types such as leucocytes and adipose tissue.

The primary HRV dependent outcome variables, 24 h SDNN and Ti, did not correlate with O3I neither did any of the other HRV parameters during dialysis, sleep-time or 24 h, following adjustment for age, BMI and use of β-blockers. Few published studies to date have assessed the relationship between LC *n*-3 PUFA status and HRV in stage 5 CKD patients on dialysis. Christensen et al. found positive correlations between granulocyte EPA, DHA and total LC *n*-3 PUFA content and 24 h SDNN in 17 patients on haemodialysis or peritoneal dialysis after a fish-oil vs. olive oil dietary intervention [17]. Svensson et al. measured LC *n*-3 PUFA in serum phospholipids in 30 patients and found no correlations with 24 h HRV measurements at baseline [18]. Rantanen et al. measured LC *n*-3 PUFA in plasma phospholipids in 135 patients and found no significant associations with 24 h time-domain or frequency-domain HRV [34].

We report that plasma EPA was significantly positively correlated with overall and longer phase components of HRV and negatively correlated with vagally-modulated beat-to-beat variability over 24 h, in contrast to the complete lack of associations between 24 h HRV and plasma DHA. The fact that a greater proportions of plasma EPA (and DPA*n*-3, the next fatty acid in the interconversion cascade from EPA to DHA) were associated with reduced beat-to-beat variability might suggest that cell membrane uptake and/or conversion of EPA to DHA is inhibited in patients on haemodialysis with lower parasympathetic tone. In fact, erythrocyte EPA to DHA ratios were significantly higher in this haemodialysis cohort compared to an age-, sex- and BMI-matched cohort of healthy participants and within the haemodialysis cohort, erythrocyte EPA:DHA ratio was inversely associated (fully adjusted) with 24 h RMSSD, HF power, LF power and SD1:SD2 ratio, all markers of vagal regulation of heartbeat [19]. Reduced availability of DHA is likely to be directly involved in impaired autonomic function by a number of mechanisms, including adverse effects on neuronal membrane properties, increasing neuroinflammation, reducing neural regeneration, and failure to inhibit oxidative stress, as well as a likely direct adverse effect on pacemaker activity [35]. These results indicate that a greater EPA proportion in plasma, is associated with better overall and longer-phase autonomic regulation but not short-term vagally-driven variability (alongside low DHA). It might be speculated that greater availability of EPA-derived lipid mediators may be involved in increased responsivity of HR to factors such as circadian fluctuations in hormone secretion and baroreflex (reflected by long-phase HRV). Reduced parasympathetic tone could be due to reduced availability of DHA in the membranes of cardiomyocytes and/or a reduction in DHA-derived lipid mediators involved in neuroprotective processes in this patient population.

This study confirms previous reports that HRV is substantially reduced in haemodialysis patients compared to healthy controls [36,37] and extends these findings by demonstrating consistently that time-domain and frequency-domain HRV parameters were significantly lower in haemodialysis patients compared to age-, sex- and BMI-matched healthy controls. Patients in this study also had SDNN measurements amongst the lowest reported in comparison to other haemodialysis cohorts [6,8,18,38]. This might be related to the fact that these were patients that recently started haemodialysis (measurements taken between 6 and 10 weeks of commencing treatment). Commencing haemodialysis is associated with autonomic dysfunction; lower SDNN values were reported in patients on haemodialysis for less than 30 months compared to those on haemodialysis for more than 30 months [37]. Moreover, after the start of haemodialysis treatments autonomic dysfunction of patients has been reported to improve or remain unaltered compared to patients with chronic renal failure before starting haemodialysis [39]. Decreases in Ti independently predict cardiac death [6], and Hayano et al. reported an independent prognostic value of Ti (<22) to predict both all-cause and sudden death (hazard ratio (95% CI); 8.1 (1.3, 48.6) and 12.6 (1.3, 126.4), respectively) [7]. In the present study 22 patients presented Ti < 22 ms which corresponds to 65% of those with valid 24 h HRV readings. Hayano et al. also reported a 24 h SDNN below 50 ms to predict a greater risk of sudden death in 31 chronic haemodialysis patients with stage 5 CKD after 5 years of follow-up [7]. Although only 2 patients presented SDNN under 50 ms in the present study, the mean 24 h SDNN of this study population was 84 ms; considered to be moderately depressed. This finding is consistent with other published studies reporting depressed HRV in haemodialysis patients [6,8,18,38,40,41,42,43]. Effects of medication on HRV should be considered in this poly-medicated patient population. For example, those taking β-blockers (n = 16) had a significantly lower HR than non-users (78 vs. 87 respectively) and higher HF (161 vs. 81), as might be expected as evidence shows that treatment with β-blockers increases HRV [44,45,46].

The inflammatory burden in CKD is amplified upon commencing haemodialysis treatment [47]. CRP has been previously positively associated with mortality in haemodialysis patients [48]. The haemodialysis patients in this study presented significantly higher hs-CRP values compared to the healthy cohort, raising the question of whether very high chronic inflammation is a possible underlying cause of low LC *n*-3 PUFA and impaired HRV. One study reported an inverse relationship between IL-6, and SDNN, SDANN and VLF in CKD patients, but in dialysis patients, no associations between HRV and inflammatory markers were found [49]. Studies with a larger sample size showed independent associations between HRV and CRP in healthy participants and in patients with CHD [50] and a meta-analysis showed that CRP predicted CHD independently of traditional cardiovascular risk factors [51].

Where erythrocyte LC *n-3* PUFA and *n*-6 PUFA contents as a proportion of total fatty acids were lower in haemodialysis patients compared to the reference group, this was met by a reciprocal increase in the proportions of MUFA and SFA in red blood cell membranes. Membrane SFA and MUFA are poor indicators of dietary fatty acid composition [52]. The fatty acids that were higher in erythrocyte membranes of the haemodialysis group–16:0, 16:1*n*-7, 18:0, 18:1*n*-7, and 18:1*n*-9–are all endogenously synthesised and some may be associated with metabolic dysregulation [53]. Higher erythrocyte membrane levels of 16:0, 16:1*n*-7, and 18:1*n*-7 were associated with greater risk of SCD in a US population-based case-control study [54]. By altering membrane fluidity and impairing potassium ion channel function, it is possible that palmitic, palmitoleic, and vaccenic acid-enrichment of cardiomyocyte membranes might also increase risk of SCD in haemodialysis patients independently of depletion in tissue LC *n*-3 PUFA and n-6 PUFA status.

This study presents several limitations, including multiple testing. However, the relationships observed were consistent across HRV parameters measuring either short-term, long-term or overall HRV making type 1 errors less likely to be a concern. There were challenges in the participant recruitment, compliance and retention of patients for the second visit. In addition, the patients enrolled in this study had different treatment schedules with patients dialysing at different times of the day (morning, afternoon or evening) which may have induced circadian differences in the readings. Another limitation of this study is the number of comorbidities presented by most patients which could have affected independently the HRV measurements not allowing the detection of potential associations with LC *n*-3 PUFA status.

Patients with stage 5 CKD are at high risk of SCD and there are numerous mechanisms that could be involved in the electrophysiological instability that predispose these patients to malignant cardiac arrhythmias. These patients have a moderate to severe cardiac autonomic dysfunction and low tissue LC *n-*3 PUFA status may contribute to increased risk of arrhythmic cardiac events and death. Research is needed to investigate the potential efficacy of EPA+DHA or DHA supplementation in this patient population for prevention of sudden cardiac death. The main finding from this feasibility study is that erythrocyte LC *n*-3 PUFA are uniformly very low in patients who have recently commenced haemodialysis, with %DHA being disproportionately lower relative to EPA. It can be concluded that other tissue biomarkers of LC *n*-3 PUFA status should be explored for their utility in assessing risk in patients undergoing haemodialysis treatment, for example, in platelets and mononuclear cells [55].

## 5. Conclusions

In summary, haemodialysis patients presented low proportions of erythrocyte membrane EPA + DHA. There were no relationships between erythrocyte membrane LC *n*-3 PUFA status and HRV, nor with estimated dietary intake of LC *n*-3 PUFA. However, negative associations between plasma EPA and erythrocyte EPA:DHA ratio, and 24 h parasympathetically driven parameters of HRV, suggest that further research is needed to investigate compromised conversion of EPA/DPA to DHA in patients that recently started haemodialysis as this may contribute to impaired cardiac autonomic function, conferring a greater risk of SCD.

## Figures and Tables

**Figure 1 nutrients-13-02453-f001:**
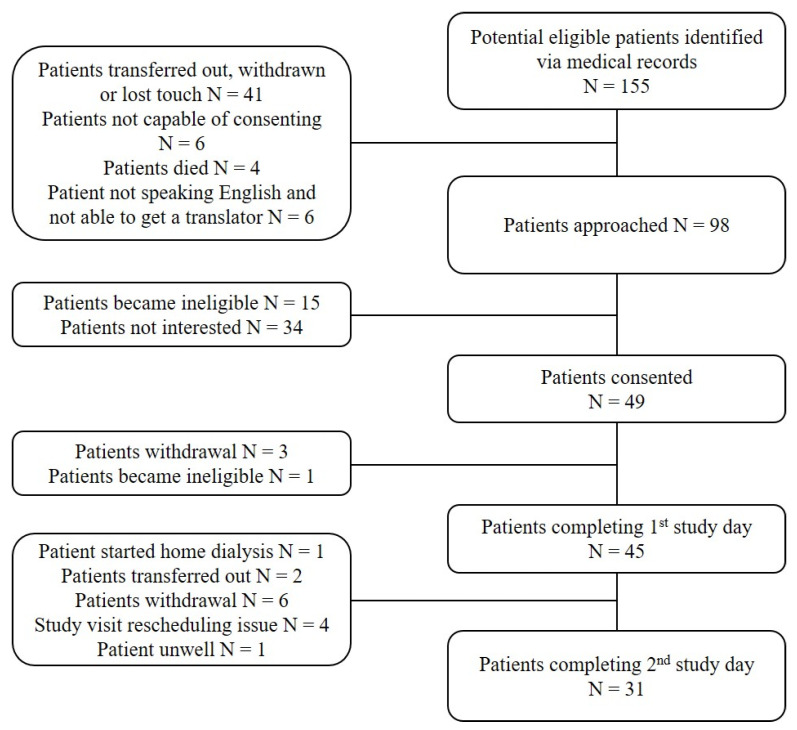
Study flow chart.

**Table 1 nutrients-13-02453-t001:** Characteristics of the study population (n = 45).

	**N**	**%**
Sex		
Male	25	56
Female	20	44
Ethnicity		
White	20	44
Black	19	42
Other	6	13
Co-morbidities		
Hypertension	40	91
Coronary artery disease	5	11
Peripheral vascular disease	2	5
Heart failure	3	7
Cerebrovascular disease	3	7
Type 2 Diabetes	17	39
Hyperlipidaemia	9	21
Anaemia	5	11
Medications		
ACE inhibitor	11	25
ARB	10	23
β-blocker	20	46
α-blocker	12	27
Diuretic	10	23
Statin	21	48
Insulin	11	25
Anti-diabetic	6	14
Anti-depressant	6	14
Vitamin D	25	57
Calcium-based phosphate binders	18	41
Iron injections	21	48
Erythropoietin therapy	20	45
Subjective daytime sleepiness		
Normal	23	74
Mild	5	16
Moderate	1	3
Severe	2	7
Risk of sleep apnoea		
High	20	65
Low	11	35
	**Mean**	**SD**
Age (years)	58	9
Waist Circumference (cm)		
Male	105.6	13.2
Female	97.1	19.4
BMI (kg/m^2^)	27.8	7.0
Average SBP (mmHg) ^1^	141	16
Average DBP (mmHg) ^1^	76	9
Fluid removal (L)	2.0	0.8
Pre- and post-dialysis measurements		
Pre-dialysis creatinine (μmol/L)	585	192
Post-dialysis creatinine (μmol/L)	228	100
Pre-dialysis albumin (g/L)		
Median	38	
Lower and upper quartiles	36, 40	
Post-dialysis albumin (g/L)		
Median	40	
Lower and upper quartiles	37, 44	
Pre-dialysis sodium (mmol/L)		
Median	139.3	
Lower and upper quartiles	137.6, 141.0	
Post-dialysis sodium (mmol/L)		
Median	138.0	
Lower and upper quartiles	137.0, 139.0	
Pre-dialysis calcium (mmol/L)	2.17	0.15
Post-dialysis calcium (mmol/L)	2.25	0.19
Pre-dialysis potassium (mmol/L)		
Geometric Mean	5.1	
95% CI	4.9, 5.4	
Post-dialysis potassium (mmol/L)		
Geometric Mean	3.8	
95% CI	3.6, 3.9	
Dialysis 2 h HR and HRV ^2^		
HR (bpm)	77.2	15.0
IBI (ms)	787	132
Ti	11.1	3.6
SDNN (ms)	45.8	13.9
pNN50 (%)		
Median	1.86	
Lower and upper quartilesRMSSD (ms)	0.68, 5.41	
Median	21.4	
Lower and upper quartiles	17.4, 36.5	
LF (ms^2^)		
Median	101	
Lower and upper quartiles	46, 197	
HF (ms^2^)		
Median	114	
Lower and upper quartiles	55, 195	
logLF:HF	−0.02	0.43
SD1:SD2		
Median	0.24	
Lower and upper quartiles	0.18, 0.42	

^1^ 12 months prior to commencing haemodialysis. ACE, angiotensin converting enzyme; ARB, angiotensin II receptor antagonists; BMI, body mass index; DBP, diastolic blood pressure; SBP, systolic blood pressure. Results expressed as percentage and numbers; mean values and standard deviations (SD); medians and interquartile rage (IQR); or geometric mean and 95% confidence intervals (CI). ^2^ Average of V1 and V2 n = 23, V1 or V2 n = 39, missing HRV values n = 6.

**Table 2 nutrients-13-02453-t002:** Comparison of LC *n*-3 PUFA status and intake, sleep-time HRV and inflammatory marker in haemodialysis patients and age-, sex- and BMI-matched healthy controls from the MARINA study.

	Haemodialysis Patients (n = 42)	Healthy Reference (n = 42)	*p-*Value
Sex (M:F)	22:20	22:20	
Age (y)	57 ± 8	57 ± 8	
Ethnicity (White:Black:Other)	19:18:5	32:7:3	0.013 ^§^
BMI (kg/m^2^)	27.6 ± 7.0	26.1 ± 3.4	0.231
hs-CRP (mg/L) ^a^	5.0 (3.3, 7.6)	0.6 (0.4, 0.9)	<0.001
EPA dietary intake (mg/day)	0.09 (0.06, 0.14)	0.10 (0.07, 0.13)	0.732
DHA dietary intake (mg/day)	0.27 (0.16, 0.45)	0.18 (0.13, 0.25)	0.176
Erythrocyte EPA (%) ^a^	0.6 (0.54, 0.77)	1.2 (1.09, 1.35)	<0.001
Erythrocyte DPA (%) ^a^	1.5 (1.37, 1.63)	3.3 (3.1, 3.5)	<0.001
Erythrocyte DHA (%)	2.6 ± 0.94	6.6 ± 1.42	<0.001
Omega-3 index (%)	3.4 ± 1.17	7.9 ± 1.67	<0.001
Erythrocyte DPA*n*-6 (%) ^a^	0.23 (0.17, 0.42)	0.45 (0.38, 0.52)	<0.001
Erythrocyte AA (%)	9.9 ± 1.29	17.2 ± 1.73	<0.001
Erythrocyte DGLA (%) ^b^	1.10 (0.88, 1.23)	1.81 (1.63, 2.07)	<0.001
Erythrocyte LA (%)	7.6 ± 1.28	11.1 ± 1.61	<0.001
Erythrocyte *cis*-VA (%) ^b^	1.41 (1.36, 1.72)	1.23 (1.11, 1.38)	<0.001
Erythrocyte OA (%)	15.7 ± 1.7	14.9 ± 1.04	<0.01
Erythrocyte SA (%)	23.1 ± 1.59	15.7 ± 1.71	<0.001
Erythrocyte POA (%) ^a^	0.42 (0.32, 0.53)	0.27 (0.23, 0.32)	<0.005
Erythrocyte PA (%) ^b^	30.4 (29.0, 31.9)	20.0 (18.8, 20.7)	<0.001
Erythrocyte EPA:DHA ratio ^b^	0.31 (0.22, 0.39)	0.18 (0.16, 0.22)	<0.001
Erythrocyte PA:LA ratio ^b^	4.00 (3.48, 4.47)	1.80 (1.65, 1.95)	<0.001
Erythrocyte AA:EPA ratio ^b^	14.0 (10.4, 19.3)	14.2 (10.7, 17.5)	0.741
Plasma EPA (%) ^a^	0.8 (0.63, 0.91)	1.0 (0.91, 1.18)	0.007
Plasma DPA (%) ^a^	0.44 (0.40, 0.47)	0.64 (0.60, 0.68)	<0.001
Plasma DHA (%)	1.8 ± 0.65	2.4 ± 0.63	<0.001
HR and HRV parameters (Sleep time)	N = 40	N = 40	
HR (bpm)	77.4 ± 12.1	64.6 ± 9.7	<0.001
IBI (ms)	800 ± 133	962 ± 115	<0.001
Ti	12.2 ± 3.8	22.6 ± 6.3	<0.001
SDNN (ms)	53.1 ± 16.7	94.3 ± 25.9	<0.001
pNN50 (%) ^a^	2.6 (1.7, 4.1)	6.2 (4.1, 9.5)	0.006
RMSSD (ms) ^a^	23.5 (19.9, 27.8)	34.4 (29.4, 40.4)	0.001
LF (ms^2^) ^a^	104 (79, 137)	655 (504, 852)	<0.001
HF (ms^2^) ^a^	106 (76, 148)	316 (231, 432)	<0.001
logLF:HF	−0.01 ± 0.82	2.64 ± 1.91	<0.001
VLF (ms^2^) ^a^	1514 (1257, 1824)	4932 (4132, 5888)	<0.001
SD1:SD2 ^a^	0.24 (0.21, 0.29)	0.20 (0.17, 0.23)	0.073

Results expressed as a ratio or mean ± SD, except ^a^ geometric means (95%CI) and *p-*value obtained by independent *t*-test, except ^§^ χ2 test, and ^b^ median (lower and upper quartiles) and *p-*value obtained by independent samples Mann-Whitney U-test. AA, arachidonic acid (20:4*n-*6); BMI, body mass index; DBP, diastolic blood pressure; DGLA, dihomo-γ-linolenic acid(20:3*n*-6); DHA, docosahexaenoic acid; DPA, docosapentaenoic acid; EPA, eicosapentaenoic acid; HR, heart rate; MA, myristic acid (14:0); OA, oleic acid (18:1*n*-9); PA, palmitic acid (16:0); POA, palmitoleic acid (16:1*n*-7); PUFA, polyunsaturated fatty acids; SA, stearic acid (18:0); SBP, systolic blood pressure; *cis*-VA, *cis*-vaccenic acid (18:1*n*-7); IBI, interbeat interval (also known as RR interval), the time interval between R spikes of the QRS complex of the electrocardiogram; HR, heart rate; bpm, beats per minute; HRV, heart rate variability; NN, normal-to-normal; Ti, triangular index (total number of all NN intervals divided by the height of the histogram of all NN intervals); SDNN, standard deviation of all NN intervals (NN intervals, similar to R-R, but on normalised IBI data); SDANN, standard deviation of the averaged NN intervals, calculated from 5min epochs; RMSSD, the square root of the mean of the sum of squares of differences between adjacent NN intervals; PNN50, percentage of adjacent NN intervals that differed by >50 ms; LF, low-frequency power; HF, high-frequency power; VLF, very-low-frequency power; SD1:SD2, the ratio of the SD of beat-to-beat IBI variability (SD1) against the SD of long-term IBI variability (SD2).

**Table 3 nutrients-13-02453-t003:** Simple and multivariable linear regression analyses between erythrocyte membrane LC *n*-3 PUFA and HRV during dialysis (2 h), sleep time and over a 24 h period.

	EPA (wt %) ^a^				DHA (wt %)				Omega-3 Index			
	*β* ^1^	*p* ^1^	*β* ^2^	*p* ^2^	*β* ^1^	*p* ^1^	*β* ^2^	*p* ^2^	*β* ^1^	*p* ^1^	*β* ^2^	*p* ^2^
**Dialysis (n = 36)**												
HR (bpm)	−0.379	0.017	−0.253	0.051	−0.251	0.123	−0.134	0.298	−0.344	0.032	−0.198	0.124
IBI (ms)	0.190	0.248	0.880	0.563	0.187	0.253	0.153	0.302	0.230	0.160	0.170	0.255
Ti	0.165	0.316	0.047	0.759	0.231	0.157	0.280	0.056	0.263	0.106	0.279	0.060
SDNN (ms)	0.136	0.408	0.023	0.882	0.101	0.542	0.142	0.347	0.133	0.419	0.141	0.354
pNN50 (%) ^a^	0.053	0.747	0.012	0.940	0.090	0.584	0.127	0.430	0.100	0.546	0.124	0.445
RMSSD (ms) ^a^	−0.099	0.549	−0.143	0.393	0.042	0.799	0.088	0.592	0.011	0.949	0.043	0.794
LF (ms^2^) ^a^	0.225	0.169	0.088	0.528	0.095	0.565	0.151	0.263	0.154	0.350	0.167	0.217
HF (ms^2^) ^a^	−0.001	0.994	−0.074	0.627	0.129	0.433	0.195	0.186	0.120	0.466	0.164	0.271
logLF:HF	0.250	0.124	0.195	0.230	−0.092	0.579	−0.101	0.528	−0.013	0.937	−0.041	0.800
SDANN (ms)	0.114	0.489	0.046	0.791	0.143	0.387	0.162	0.339	0.166	0.313	0.166	0.329
SD1:SD2 ^a^	−0.201	0.220	−0.179	0.305	−0.029	0.863	−0.002	0.993	−0.085	0.608	−0.052	0.766
**Sleep time (n = 39)**												
HR (bpm)	−0.306	0.062	−0.274	0.110	−0.086	0.608	−0.034	0.840	−0.169	0.304	−0.110	0.522
IBI (ms)	0.309	0.059	0.270	0.114	0.079	0.637	0.032	0.850	0.154	0.349	0.106	0.532
Ti	0.184	0.268	0.128	0.469	−0.018	0.915	−0.066	0.701	0.037	0.825	−0.027	0.879
SDNN (ms)	0.220	0.185	0.156	0.378	−0.035	0.836	−0.066	0.703	0.032	0.850	−0.018	0.916
pNN50 (%) ^a^	−0.043	0.798	−0.135	0.429	0.180	0.280	0.162	0.341	0.114	0.496	0.033	0.846
RMSSD (ms) ^a^	−0.043	0.797	−0.127	0.466	0.173	0.300	0.132	0.435	0.148	0.376	0.089	0.602
LF (ms^2^) ^a^	0.156	0.349	0.018	0.909	0.164	0.327	0.157	0.309	0.197	0.235	0.153	0.323
HF (ms^2^) ^a^	0.096	0.568	−0.013	0.938	0.230	0.166	0.202	0.207	0.241	0.145	0.187	0.247
logLF:HF	0.055	0.743	0.022	0.902	−0.099	0.556	−0.073	0.674	−0.076	0.652	−0.063	0.719
VLF (ms^2^) ^a^	0.268	0.104	0.194	0.266	−0.101	0.548	−0.137	0.421	0.018	0.913	−0.035	0.844
SDANN (ms) ^a^	0.145	0.377	0.203	0.221	−0.127	0.446	−0.161	0.361	−0.059	0.725	−0.099	0.580
SD1:SD2 ^a^	−0.200	0.229	−0.239	0.177	0.214	0.197	0.193	0.267	0.141	0.399	0.115	0.515
**24 h (n = 35)**												
HR (bpm)	−0.255	0.145	−0.152	0.451	−0.216	0.220	−0.146	0.430	−0.279	0.110	−0.200	0.308
IBI (ms)	0.315	0.069	0.213	0.282	0.217	0.217	0.148	0.414	0.297	0.088	0.219	0.256
Ti ^a^	0.430	0.011	0.334	0.094	−0.008	0.965	−0.078	0.677	0.109	0.538	0.001	0.998
SDNN (ms)	0.392	0.022	0.210	0.289	−0.035	0.842	−0.173	0.342	0.073	0.684	−0.136	0.484
pNN50 (%)	−0.051	0.775	−0.145	0.508	0.164	0.355	0.186	0.351	0.145	0.414	0.168	0.432
RMSSD (ms)	−0.230	0.190	−0.330	0.130	0.191	0.279	0.256	0.201	0.123	0.489	0.197	0.359
LF (ms^2^) ^a^	0.173	0.329	−0.175	0.296	0.265	0.130	0.221	0.145	0.304	0.081	0.198	0.222
HF (ms^2^) ^a^	−0.090	0.611	−0.311	0.112	0.286	0.102	0.323	0.068	0.252	0.150	0.276	0.149
logLF:HF	0.300	0.085	0.127	0.503	0.033	0.851	−0.047	0.786	0.114	0.521	−0.019	0.918
VLF (ms^2^)	0.412	0.016	0.293	0.150	−0.050	0.777	−0.168	0.373	0.063	0.722	−0.109	0.588
SDANN (ms)	0.395	0.021	0.239	0.239	−0.067	0.707	−0.211	0.255	0.043	0.810	−0.171	0.390
SD1:SD2 ^a^	−0.349	0.043	−0.310	0.146	0.196	0.267	0.316	0.102	0.095	0.594	0.269	0.197

*β*^1^*—*unadjusted standardised beta coefficient; *p*^1^—unadjusted *p-*value; *β*^2^*—*adjusted standardised beta coefficient; *p*^2^ value, adjusted for age, BMI and *β*-blockers (yes or no). 24 h HRV analysis has activity level from accelerometery data (counts per minute, cpm) as an additional adjustment for *p*^2^. VLF during dialysis not presented due to shorter duration of recording (2 h). ^a^ Regression analysis based on LN variables. IBI, interbeat interval (also known as RR interval), the time interval between R spikes of the QRS complex of the electrocardiogram; HR, heart rate; bpm, beats per minute; HRV, heart rate variability; NN, normal-to-normal; Ti, triangular index (total number of all NN intervals divided by the height of the histogram of all NN intervals); SDNN, standard deviation of all NN intervals (NN intervals, similar to R-R, but on normalised IBI data); SDANN, standard deviation of the averaged NN intervals, calculated from 5 min epochs; RMSSD, the square root of the mean of the sum of squares of differences between adjacent NN intervals; PNN50, percentage of adjacent NN intervals that differed by >50 ms; LF, low-frequency power; HF, high-frequency power; VLF, very-low-frequency power; SD1:SD2, the ratio of the SD of beat-to-beat IBI variability (SD1) against the SD of long-term IBI variability (SD2).

**Table 4 nutrients-13-02453-t004:** Simple and multivariable linear regression analyses between plasma LC *n*-3 PUFA and HRV during dialysis (2 h), sleep time and over a 24 h period.

	EPA ^a^				DPA(*n*-3) ^a^				DHA			
	*β* ^1^	*p* ^1^	*β* ^2^	*p* ^2^	*β* ^1^	*p* ^1^	*β* ^2^	*p* ^2^	*β* ^1^	*p* ^1^	*β* ^2^	*p* ^2^
**Dialysis (n = 39)**												
HR (bpm)	−0.129	0.435	−0.119	0.347	0.207	0.207	0.145	0.253	−0.298	0.065	−0.208	0.101
IBI (ms)	0.151	0.359	0.159	0.273	−0.178	0.279	−0.111	0.448	0.239	0.143	0.223	0.128
Ti	0.143	0.386	0.134	0.362	−0.051	0.759	−0.047	0.752	0.328	0.042	0.331	0.022
SDNN (ms)	0.123	0.454	0.118	0.422	−0.100	0.546	−0.087	0.558	0.252	0.122	0.258	0.081
pNN50 (%) ^a^	−0.067	0.686	−0.052	0.743	−0.159	0.334	−0.118	0.459	0.056	0.737	0.092	0.567
RMSSD (ms)	−0.196	0.231	−0.184	0.249	−0.279	0.085	−0.253	0.112	−0.023	0.892	0.015	0.925
LF (ms^2^) ^a^	0.095	0.564	0.081	0.539	−0.147	0.372	−0.153	0.248	0.239	0.142	0.235	0.076
HF (ms^2^) ^a^	−0.086	0.602	−0.073	0.617	−0.221	0.176	−0.185	0.202	0.120	0.468	0.168	0.254
logLF:HF	0.196	0.231	0.166	0.285	0.095	0.566	0.042	0.791	0.100	0.546	0.044	0.781
SDANN (ms)	0.182	0.266	0.177	0.282	−0.013	0.939	−0.007	0.965	0.270	0.096	0.270	0.105
SD1:SD2 ^a^	−0.308	0.057	−0.291	0.076	−0.231	0.158	−0.211	0.209	−0.202	0.218	−0.163	0.337
**Sleep time (n = 40)**												
HR (bpm)	−0.263	0.106	−0.279	0.080	0.131	0.427	0.080	0.626	−0.225	0.168	−0.205	0.210
IBI (ms)	0.280	0.085	0.296	0.060	−0.100	0.543	−0.046	0.780	0.218	0.183	0.201	0.218
Ti	0.191	0.244	0.186	0.264	−0.146	0.375	−0.133	0.431	0.127	0.441	0.090	0.599
SDNN (ms)	0.254	0.119	0.246	0.137	0.040	0.808	0.052	0.758	0.046	0.781	0.006	0.970
pNN50 (%) ^a^	−0.180	0.274	−0.174	0.272	−0.362	0.024	−0.321	0.041	−0.113	0.495	−0.164	0.309
RMSSD (ms)	−0.183	0.264	−0.176	0.277	−0.331	0.040	−0.290	0.071	−0.116	0.482	−0.153	0.353
LF (ms^2^) ^a^	0.031	0.851	0.025	0.869	−0.142	0.390	−0.110	0.479	0.068	0.680	0.035	0.823
HF (ms^2^) ^a^	−0.048	0.773	−0.042	0.789	−0.222	0.174	−0.173	0.268	−0.015	0.929	−0.047	0.765
logLF:HF	0.088	0.593	0.075	0.658	0.098	0.553	0.079	0.644	0.086	0.604	0.087	0.616
VLF (ms^2^) ^a^	0.285	0.079	0.277	0.094	0.078	0.639	0.085	0.610	0.099	0.551	0.053	0.754
SDANN (ms) ^a^	0.320	0.047	0.308	0.064	0.181	0.270	0.172	0.316	0.039	0.812	0.004	0.980
SD1:SD2 ^a^	−0.354	0.027	−0.340	0.036	−0.364	0.023	−0.330	0.045	−0.159	0.333	−0.172	0.311
**24 h (n = 35)**												
HR (bpm)	−0.280	0.109	−0.223	0.195	0.196	0.266	0.154	0.369	−0.298	0.087	−0.256	0.170
IBI (ms)	0.352	0.041	0.296	0.077	−0.144	0.416	−0.097	0.568	0.335	0.053	0.297	0.103
Ti ^a^	0.347	0.044	0.281	0.112	−0.222	0.206	−0.211	0.221	0.282	0.107	0.221	0.244
SDNN (ms)	0.441	0.009	0.378	0.022	−0.052	0.769	−0.026	0.879	0.254	0.147	0.124	0.506
pNN50 (%)	−0.302	0.083	−0.331	0.073	−0.394	0.021	−0.376	0.038	−0.075	0.675	−0.097	0.636
RMSSD (ms)	−0.337	0.052	−0.353	0.057	−0.387	0.024	−0.386	0.034	−0.120	0.498	−0.117	0.573
LF (ms^2^) ^a^	0.062	0.729	−0.017	0.909	−0.134	0.449	−0.108	0.451	0.219	0.213	0.089	0.574
HF (ms^2^) ^a^	−0.211	0.232	−0.259	0.125	−0.343	0.047	−0.310	0.061	0.031	0.863	0.003	0.986
logLF:HF	0.298	0.087	0.258	0.106	0.210	0.234	0.205	0.199	0.240	0.171	0.125	0.481
SDANN (ms)	0.464	0.006	0.405	0.016	−0.016	0.929	−0.010	0.954	0.247	0.159	0.121	0.525
VLF (ms^2^)	0.481	0.004	0.434	0.010	−0.038	0.829	−0.010	0.956	0.249	0.155	0.155	0.422
SD1:SD2 ^a^	−0.482	0.004	−0.461	0.009	−0.331	0.056	−0.337	0.060	−0.227	0.197	−0.152	0.453

*β*^1^—unadjusted standardised beta coefficient; *p*^1^—unadjusted *p-*value; *β*^2^—adjusted standardised beta coefficient; *p*^2^ value, adjusted for age, BMI and *β*-blockers (yes or no). 24 h HRV analysis has activity level from accelerometery data (counts per minute, cpm) as an additional adjustment for *p*^2^. VLF during dialysis not presented due to shorter duration of recording (2 h). ^a^ Regression analysis based on LN variables. IBI, interbeat interval (also known as RR interval), the time interval between R spikes of the QRS complex of the electrocardiogram; HR, heart rate; bpm, beats per minute; HRV, heart rate variability; NN, normal-to-normal; Ti, triangular index (total number of all NN intervals divided by the height of the histogram of all NN intervals); SDNN, standard deviation of all NN intervals (NN intervals, similar to R-R, but on normalised IBI data); SDANN, standard deviation of the averaged NN intervals, calculated from 5 min epochs; RMSSD, the square root of the mean of the sum of squares of differences between adjacent NN intervals; PNN50, percentage of adjacent NN intervals that differed by >50 ms; LF, low-frequency power; HF, high-frequency power; VLF, very-low-frequency power; SD1:SD2, the ratio of the SD of beat-to-beat IBI variability (SD1) against the SD of long-term IBI variability (SD2).

## Data Availability

The data presented in this study are available on request from the corresponding author.

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
