# Peer review of "Heart Rate Variability and Long Chain n-3 Polyunsaturated Fatty Acids in Chronic Kidney Disease Patients on Haemodialysis: A Cross-Sectional Pilot Study"

_nutrients, 2021, doi:10.3390/nu13072453_

Round 1

Reviewer 1 Report

The study by Pinto et al. entitled “Heart rate variability and long chain n-3 polyunsaturated fatty acids in chronic kidney disease patients on hemodialysis: a 3 cross-sectional pilot study” shows that hemodialysis patients have altered level of erythrocyte eicosatetraenoic acid and docosahexaenoic acid which correlated with heart rate variability during 24h observation. The study is interesting and presents relevant data. I have some minor comments listed below that should be addressed by the authors.

Methods:

The authors repeat information about measured parameters in blood samples: lines 104 to 106 and lines 141 to 143. Please verify and correct it if necessary.

The authors should also discuss and stress the potential link observed between heart rate variability and long chain n-3 polyunsaturated fatty acids content in analyzed disease. How these potentially unrelated factors could affect each other especially in CKD.

Reviewer 2 Report

The authors have investigated the association of Long chain n-3 polyunsaturated fatty acids in heart rate variability in dialysis, sleep and 24 h in patients who have recently commenced haemodialysis.

The work is well conducted; however, the following points need to be addressed:

1) To write LC n-3 PUFA in similar manner (es. line 51 and 52 is different)

2) In line 201 maybe “oily” is not correct?

3) In these patients there are a metabolic alteration and the authors in paragraph “3.3.2. Plasma LC n-3 PUFA” considered only the changes of LC n-3 PUFA and I think that it is important all fatty acid profile and also to analyse and to discuss the changes in n6 PUFA as arachidonic acid or saturated fatty acids.

4) In discussion the authors report “Background diet may play a role but this was not supported by food frequency questionnaire data on self-reported fish intake. It is most likely that the major contributing factor to low erythrocyte LC n-3 PUFA content was related to the pathology of kidney disease” and it is necessary to expand the data on diet and to report in a table the number of erythrocytes present in these patients.

Reviewer 3 Report

Article - ,, Heart rate variability and long chain n-3 polyunsaturated fatty 2 acids in chronic kidney disease patients on haemodialysis: a 3 cross-sectional pilot study "- presents LC n-3 PUFA status in patients who recently commenced haemodialysis, and to investigate relationships between erythrocyte membrane LC n-3 PUFA composition and HRV in these patients,
including comparison to a reference healthy population. The research design presented in the paper is appropriate and the results obtained are clearly presented. As a suggestion for the authors would be to conduct a study supplementing patients diet with omega-3 PUFA dietary supplements.

Round 2

Reviewer 1 Report

I accept the manuscript in the present form.

Author Response

Thank you for your review and approval.

Reviewer 2 Report

Thank you for your modifications in the manuscript but it is still necessary to change the manuscript. I reported my comments in blue after yours responses.
